# Science and Technology of <u>Hi</u>gh <u>Per</u>formance <u>Fer</u>ritic (HiperFer) Stainless Steels

**Bernd Kuhn [1],\*****, Michal Talik [1],†, Torsten Fischer [1]****, Xiuru Fan [1], Yukinori Yamamoto [2] and Jennifer Lopez Barrilao [1],‡**

1   Institute of Energy and Climate Research (IEK), Microstructure and Properties of Materials (IEK-2), Forschungszentrum Juelich GmbH, 52425 Jülich, Germany; michal.talik@voestalpine.com (M.T.); t.fischer@fz-juelich.de (T.F.); x.fan@fz-juelich.de (X.F.); je.lopez@fz-juelich.de (J.L.B.)

2   Materials Science and Technology Division, Oak Ridge National Laboratory, Oak Ridge, TN 37831-6115, USA; y.yamamoto@ornl.gov

\*   Correspondence: b.kuhn@fz-juelich.de; Tel.: +49-2461-61-4132

†   Current address: Voestalpine Böhler Welding UTP Maintenance GmbH, Elsäßer Str. 10, 79189 Bad Krozingen, Germany.

‡   Jennifer Lopez Barrilao, Independent Researcher, 52425 Jülich, Germany.

**Abstract:** Future, flexible thermal energy conversion systems require new, demand-optimized high-performance materials. The <u>Hi</u>gh performance <u>Fer</u>ritic (HiperFer) stainless steels, under development at the Institute of Microstructure and Properties of Materials (IEK-2) at Forschungszentrum Jülich GmbH in Germany, provide a balanced combination of fatigue, creep and corrosion resistance at reasonable price. This paper outlines the scientific background of alloy performance development, which resulted in an age-hardening ferritic, stainless steel grade. Furthermore, technological properties are addressed and the potential concerning application is estimated by benchmarking versus conventional state of the art materials.

**Keywords:** HiperFer; fatigue; creep; reactive strengthening; laves phase

## 1. Introduction

The German "Energiewende" poses demanding challenges with regard to the development, operation and maintenance of flexible, regenerative energy converters and storage systems (e.g., pumped thermal electricity storage [1,2], concentrating solar power [3,4], biomass firing, power-to-X technologies [5], etc.) and conventional back-up power plants. At present, development is focused on process related issues, with development of new materials, suitable to meet future requirements, not playing a major role. The knowledge of cyclic, microstructural damage and its effect on the failure mechanisms and potentially associated loss of lifetime of conventional heat resistant, structural materials is in need of improvement. The development of new materials, optimized for cyclic operation, suffers from this shortcoming.

So-called <u>a</u>dvanced <u>f</u>erritic-<u>m</u>artensitic (AFM) 9–12 wt.% Cr steels, which feature tempered martensite structure and offer creep strength and corrosion resistance up to application temperatures of 600 to 620 °C [6,7], are typical, (low-cost) structural materials for ultra-supercritical steam power plants. Because of limited steam oxidation resistance the 9 wt.% Cr materials cannot be applied beyond 620 °C [7]. Improved 12 wt.% chromium AFM steels were developed, but do exhibit a sigmoidal decrease in creep strength [8], caused by the formation of the so-called Z-phase (a complex Cr(V,Nb)N compound) at the expense of strengthening MX particles [9,10] during long-term application. However, an increase in chromium content is considered essential to ensure sufficient steam oxidation

resistance [11,12] up to temperatures of 650 °C. Furthermore, 9–12 Cr steels are not resistant to downtime corrosion, which is one of the main reasons for a significant increase in expenses for conservation of flexibly operated German power plants [13]. In the light of the quite complex alloy composition of AFM steels and the multitude of new requirements from future power engineering, the further development of this alloy class seems to be left at an irresolvable conflict of aims.

Novel high chromium <u>High</u> performance <u>Fer</u>ritic (HiperFer) [14] steels, developed by Forschungszentrum Jülich GmbH, Germany provide a promising way out of this technological dead end. Strengthening of ferritic stainless steel cannot be accomplished by MX particles, because of the lacking C and N solubility in the fully ferritic matrix. In contrast, alloying by suitable amounts of niobium and tungsten ensures a combination of solid solution and intermetallic $(Fe,Cr,Si)_2(Nb,W)$ Laves particle strengthening, which enables creep strength potential beyond grade 92 [14–16] and steam oxidation resistance superior to 12 wt.% Cr AFM steels [16]. Operational flexibility will strongly grow in importance in future thermal power conversion [13]. For this reason, increased thermomechanical fatigue resistance [14,17,18] was the main focus of HiperFer development, with creep strength as a subordinate, but still relevant issue. HiperFer steel is fully ferritic, without martensitic re-transformation in the welding cycle, and for this reason intrinsically free from so-called TypeIV (i.e., fine-grain heat-affected zone) cracking.

## 2. Materials and Methods

### 2.1. Alloy Design

Having low solubility [19,20] and comparably high diffusion rate [21,22] in ferrite, Nb is a key element in the design of ferritic, stainless, intermetallic particle strengthened steel. It is a strong Laves-phase former [23] and in combined alloying with tungsten, for improved solid solution strengthening, and silicon to accelerate nucleation [24–28], forms a thermodynamically stable $(Fe,Cr,Si)_2(Nb,W)$-Laves phase [16,29,30]. With Nb being a strong carbonitride former it is necessary to restrict C and N to a minimum (< 0.01 wt.%), because even small contents of these species may decrease the amount of Nb available [28,31–33] for Laves phase precipitation. Furthermore, primary TiN particles may act as nucleation sites of niobium consuming Nb(C, N) [34,35], which may additionally affect the formation and stability of the desired Laves phase particles in a negative way. The implications for the design of Laves phase strengthened, ferritic, stainless steels for structural applications are covered in detail in [16,36].

The aims of alloy development are manifold: First, maximization of the amount of strengthening Laves phase was desired to increase fatigue and creep strength. Second, the content of the $(Fe,Cr)$-σ-phase, which usually is considered to deteriorate ductility, hot-workability and weldability, corrosion resistance and long-term thermomechanical fatigue properties [37–40], should be restricted to a minimum (favorably below 600 °C). Third, just one single strengthening intermetallic phase—the Laves phase—was sought (i.e., precipitation of e.g., χ $(Fe_{36}Cr_{12}W_{10})$ and μ $(Fe_7W_6)$ phase should be avoided) in the envisaged operation temperature range from 580 to 650 °C to facilitate welding and long-term stability of microstructure. Spinodal decomposition, i.e., nucleation of a second, Cr-rich ferrite phase (cf. Figure 1: α-Cr) at low temperature, should be minimized and improvement of mechanical properties should not be accomplished on the expense of corrosion and steam oxidation resistance in comparison to the 22 Cr model steels covered in [16].

Changes in chemical composition do not only directly affect application properties, but also the temperature window and characteristics of processing (cf. Figure 1). Materials suitable for common processing (of e.g., tubes, pipes, plates, sheets, forgings, etc.) temperatures of stainless steels (950–1200 °C) are favored. Regarding this multitude of prerequisites, thermodynamic equilibrium calculations were carried out, utilizing the software package Thermo-Calc® (applying database version TCFE7) for the nominal compositions of both the trial alloy types.

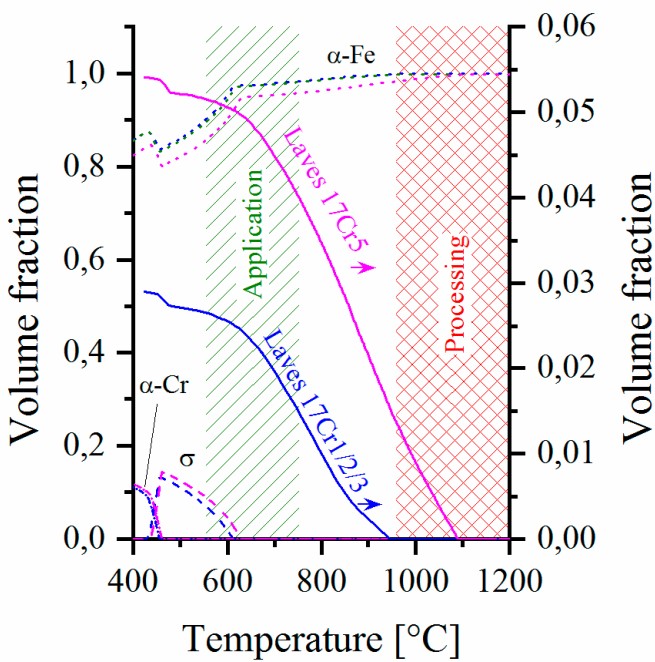

**Figure 1.** Simplified phase diagrams of the 17Cr1/2 "workhorse" (nominal: 17 Cr, 2.6 W, 0.6 Nb, 0.25 Si) and the advanced 17Cr5 (nominal: 17 Cr, 4 W, 1 Nb, 0.25 Si) trial steels (Thermo-Calc®, TCFE7).

According to these the lower alloyed "workhorse" 17Cr1/2 steel (nominal: 17 Cr, 2.6 W, 0.6 Nb, 0.25 Si) contains a volume fraction of about 2.23 vol.% of Laves phase at 650 °C. The increased W- and Nb-contents of the advanced 17Cr5 (nominal: 17 Cr, 4 W, 1 Nb, 0.25 Si) composition yield a more than doubled volume fraction of 4.82 vol.%. According to the calculation, both steels contain small contents of the σ- and α-Cr- phases below 610 and 450 °C, respectively.

While it is known that thermodynamic calculation overestimates the σ-phase range in this alloying system [16], no practical information on the calculation accuracy is available concerning the α-Cr-phase, as far as the authors are aware of. The calculated chromium concentration of the ferrite matrix does not drop below 16.5 wt.% in the application temperature range and thus ensures sufficient steam oxidation [11] and downtime corrosion [41,42] resistance.

The two trial alloy types are characterized by a fundamental difference: The low-alloyed 17Cr1/2 workhorse composition represents an alloy, similar to (but not matching) the philosophy of current AFM steels, which need multi-step quality heat treatment to reach optimum strength. In the case of AFM steel, this consists of austenitizing with martensitic transformation during subsequent rapid cooling and a second (or even third) holding step at temperatures (comparatively far) above the envisaged application temperature for stress relief and precipitation heat treatment. With kinetics being tuned to rapid precipitation HiperFer 17Cr1/2 can either be put into service in the cold-rolled or recrystallized + precipitation heat treated state, or precipitation heat treatment could potentially be executed during plant commissioning (outlined in Section 2.4). Standard quality heat treatment above application temperature thus becomes obsolete. This batch served as the "workhorse" to study the fundamental interactions of chemistry, processing, precipitation, resulting microstructure and mechanical properties. The gained know-how then was transferred to the advanced 17Cr5 variant, which is designed for age-hardening and thus redundantizes any standard quality and even simplified precipitation heat treatment (additional to recrystallization for grain size adjustment) after the forming process. The increase in tungsten content reduces primary creep strain by increased solution strengthening, while the higher content of niobium boosts precipitation kinetics and thus minimizes accumulation of creep strain during decomposition of the supersaturated solid solution and nucleation of the strengthening Laves phase precipitates in the early primary creep stage.

*2.2. Base Material Production, Processing and Picrostructure*

The model steels were produced by the Steel Institute (IEHK) of the Northrhine-Westfalian Technical University Aachen (RWTH), Germany from high purity raw materials by vacuum induction melting of 80 kg ingots and casting to original block dimensions of 140 mm × 140 mm × ~ 525 mm. The blocks were then forged to 80 mm × 56 mm, air cooled and cut into pieces of 135 mm in length. Subsequently soaking was carried out at 1140–1180 °C for 2 h. The 17Cr1 blocks were hot-rolled to a final plate thickness of 15.5 mm at 1000 °C and subsequently air-cooled, while the 17Cr2 material was cold-rolled 920 °C and subsequently water quenched.

Hot-rolling (referred to as "HR" in the following text) above the dissolution temperature of the Laves phase in combination with comparably slow air cooling leads to almost globular grain morphology with low dislocation density and few, small Laves phase particles, mainly located at high angle grain boundaries (Figure 2a). In contrast, cold-rolling ("CR" in the following) below the dissolution temperature of the Laves phase, followed by rapid water quenching, results in a typical deformation morphology with elongated grains (Figure 2b), high dislocation density and rarely any Laves phase particles. Globular grain morphology can be obtained by recrystallization annealing above the Laves phase dissolution temperature. To obtain equi-axed, globular grain structure and full dissolution of Laves phase particles (originating from preceding forming), the rolled 17Cr2 plates were annealed at 1025–1050 °C for 15 min and subsequently water quenched. 1100–1125 °C for 25 min were applied in case of the 17Cr5 plates.

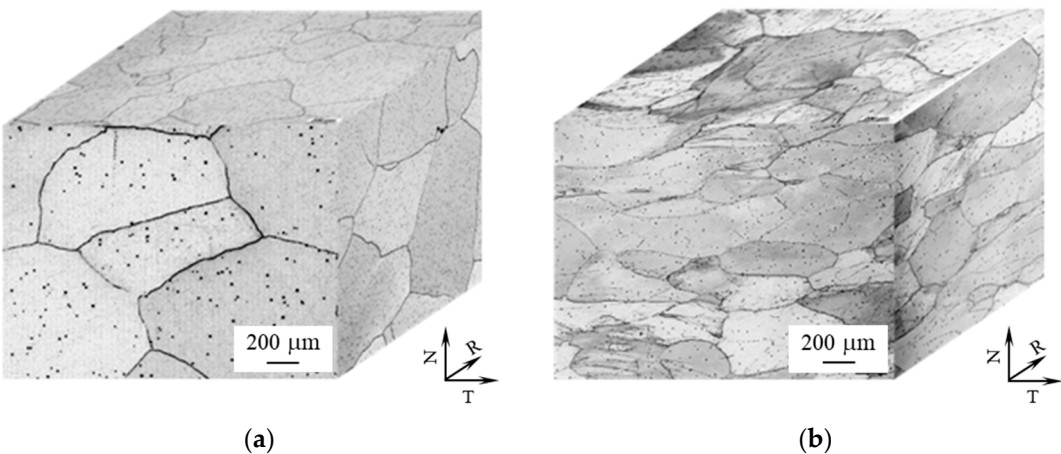

(**a**)                                                                 (**b**)

**Figure 2.** Typical microstructures of (**a**) hot-rolled 17Cr1 and (**b**) cold-rolled 17Cr2 HiperFer steel (R: Rolling, T: Transversal, N: Normal direction of rolled plate material; obtained from optical micrographs taken from the three directions).

The chemical compositions of the trial steels (analyzed by Inductively Coupled Plasma Optical Emission Spectroscopy (ICP-OES); C, N analyzed by infrared absorption) and commercial 316L, grade 92, MarBN (from original material certificates) are given in Table 1.

Commercial grade 92 (1040–1070 °C, 2 h / 730–800 °C, 2 h) material was supplied in the form of billet (diameter: 200 mm), MarBN (1120 °C/1 h/air cooling + 700 °C/2 h/air cooling + 700 °C/4 h/air cooling) in slab form of 100 mm × 200 mm × 68 mm dimension and 316L as sheet product of 16 mm thickness.

**Table 1.** Chemical composition (wt.-%) of the ferritic trial alloys and benchmark steels.

| Batch-ID: | C | N | Cr | Mn | Si | Nb | W | V | Al | Ni | Mo | B |
|---|---|---|---|---|---|---|---|---|---|---|---|---|
| **HiperFer 17Cr1** | <0.01 | <0.01 | 16.7 | 0.46 | 0.23 | 0.56 | 2.42 | - | - | - | - | - |
| **HiperFer 17Cr2** | <0.01 | <0.01 | 17.1 | 0.18 | 0.25 | 0.63 | 2.41 | - | - | - | - | - |
| **HiperFer 17Cr5** | <0.01 | <0.01 | 17.2 | 0.20 | 0.27 | 0.99 | 3.70 | - | - | - | - | - |
| **316L** | 0.024 | 0,07 | 16.85 | 1.35 | 0.44 | - | - | - | - | 10.00 | 2.06 | - |
| **Grade 92** | 0.16 | 0.051 | 8.96 | 0.46 | 0.04 | 0.069 | 1.84 | 0.2 | 0.007 | 0.06 | 0.47 | 0.001 |
| | | | | | | **Nd** | | | | | **Co** | |
| **MarBN** | 0.09 | 0.009 | 8.97 | 0.48 | 0.25 | 0.03 | 3.0 | 0.2 | 0.01 | 0.09 | 3.0 | 0.011 |

*2.3. Trial Welds*

First trial welds of as-rolled plate material (machined to 12 mm in thickness) were produced by gas tungsten arc welding (GTAW) at the Oak Ridge National Laboratory Materials Science and Technology Division, applying compositionally matching metal strips (1.6 mm diameter), taken from the base metal plates. A single V-shape groove was filled with compositionally matched weld filler metal by eleven weld beads, which were applied in a pre-heat and inter-pass temperature range from 120 to 150 °C. Visual inspection of the weld surface and cross-sectional observation did not reflect welding related defects and the welds successfully passed the ASTM E190-14 [43] side bend test. These welds intentionally represent the worst-case scenario of a high dislocation density (i.e., as-rolled, forged, bent) material, being welded and later on put into service without conventional post-weld heat treatment. Microstructural changes, i.e., recovery of excess dislocations or partly recrystallization and consequently a drop in dislocation strengthening within the heat-affected zone, are inevitable under these prerequisites. Details on weld microstructure can be found in [44].

*2.4. Heat Treatment*

HiperFer is designed to stay fully ferritic at all temperatures to keep it free from fine grain formation in the heat-affected zones of welds [45,46]. For this reason the mechanical properties of this type of steel are highly dependable on processing (i.e., rolling, forging, bending, welding), because it does not undergo martensitic transformation even during rapid cooling, which is implemented to prevent uncontrolled formation of precipitates, from process heat. This results in comparatively low dislocation density (i.e., less nucleation sites for strengthening precipitates) and thus comparably low, expected mechanical strength in the as-processed state. This could be counterbalanced by tailored thermomechanical processing [44] like cold-rolling. Mechanical properties depending on thermomechanical processing might restrict utilization of the proposed steels to applications, where component production and plant construction do not cause significant microstructural alterations, e.g., to components, which are not welded. Furthermore, implied complexity and additional cost might prohibit market entry or deeper market penetration.

This drawback is resolved by tailored precipitation kinetics, which enable simplified, short-term precipitation annealing (PA) in the envisaged application temperature range. Potentially, even precipitation heat treatment during plant commissioning would be feasible. Annealing for 0.5 to 10 h in the temperature range from 600 to 650 °C, followed by water quenching is effective in increasing the mechanical properties of hot-rolled and restoring the properties of recrystallization annealed HiperFer material. A typical resulting microstructure is depicted in Section 3.1 (cf. Figure 4). Intra- and intergranular precipitation of small Laves phase particles effectively strengthens the material. High angle grain boundaries are characterized by alongside particle-free zone (PFZ) formation, which can be influenced by alloy composition and heat-treatment. Engineering these PFZs is important in controlling creep ductility (cf. [47] and Section 3.4.2, Figure 11).

*2.5. Mechanical Testing*

For tensile, creep and relaxation testing cylindrical specimens with gauge diameters of 6.4 mm and gauge lengths of 30 mm were applied. All samples for mechanical testing were taken from the plate materials perpendicular to the rolling direction.

The tensile experiments were performed at strain rates of $10^{-3}s^{-1}$ at ambient (according to DIN EN 10002-1) and $8.33 \cdot 10^{-5}s^{-1}/8.33 \cdot 10^{-4}s^{-1}$ (in the elastic/plastic range, according to DIN EN 10002-5) at elevated temperatures, utilizing an Instron (Norwood, MA, USA) Type 1362 testing machine with 10 kN of load capability.

Creep experiments were carried out in single specimen, constant load, lever-arm type creep machines, with continuous elongation measurement at the gauge portions of the specimens. In case of specimens tested at stress levels below 100 MPa only the primary creep stages were recorded in single specimen machines. Upon entering the secondary stage of creep the samples were transferred to multi-specimen machines for long-term testing (unless otherwise stated) in discontinuous creep experiments with periodical measurement of specimen strain. Optical strain measurements were carried out after cooling to ambient temperature, unloading and taking the specimens from the testing equipment for length measurement. This procedure was repeated in nominally 1000 h intervals until rupture. Electrical three-zone furnaces, controlled to the specified testing temperatures with an accuracy of +/− 2 °C, were used in the tensile and single specimen creep experiments. The multi-specimen machines for discontinuous creep testing were equipped with electrical five-zone furnaces, controlled to an accuracy of +/− 3 °C.

Relaxation experiments were executed utilizing an Instron Type 1362 testing machine with 10 kN of load capability. At testing temperature, the specimens were loaded to the initial stress level (250 MPa @ 600 and 200 MPa @ 650 °C), applying a controlled strain rate of $10^{-3}s^{-1}$. The relaxation then was initiated by switching the machine to strain control. While the strain level reached after loading was kept constant stress relaxation was recorded.

Type R (Pt/RhPt) thermocouples were attached to the specimen gauge lengths for temperature control in tensile tests and continuous creep testing, while shielded (Type R) temperature measuring rods, in close vicinity to the specimens, were utilized in interrupted creep testing. In tensile, creep and relaxation testing all specimens were heated to the designated testing temperature at a rate of 5 K/min. and maintained for about one hour before starting the testing machine or applying the load for ensuring thermal equilibrium conditions.

According to the European Code-of-Practice [48] strain controlled thermomechanical fatigue (TMF) testing was executed at cylindrical specimens with a gauge length of 15 mm and a diameter of 7 mm, utilizing servo-hydraulic fatigue testing systems with inductive specimen heating. Type R sling thermocouples were utilized to control temperature. A so called "out-of-phase cycle (oop)" was started by heating from the minimum temperature of 50 °C to 650 °C maximum temperature at a controlled rate of 10 Ks$^{-1}$. After reaching the maximum temperature the specimen was immediately cooled down at the same rate (10 Ks$^{-1}$) by compressed air, i.e., no holding times were implemented into the cycle, to restrict creep to a minimum. During heating and cooling the thermal expansion of the material was either fully (100% oop) or partly (80, 60, 45% oop) obstructed by the testing machine. Below ~ 150 °C, cooling was retarded due to insufficient amounts of cooling air. For this reason, the duration of the cooling cycle was 85 s. The duration of the whole cycle was 145 s.

Fatigue crack growth (FCG) experiments at compact tension (CT) specimens were accomplished utilizing a servo-hydraulic Instron (Norwood, Massachusetts, USA) Model 1343 testing machine with inductive heating. A modified specimen geometry (width, W: 40 mm, thickness, B: 10 mm, machined notch depth, $a_n$: 10 mm) was utilized, because of limited material availability. The dimensions are in accordance with the ASTM fatigue crack growth testing standard [49]. Pre-cracking of the specimens up to a starter crack length $a_0$ to width ratio (a/W) of 0.4 was performed at ambient temperature in an Instron Model 1603 resonance tester. The direct current potential drop (PD) technique was employed to record crack length during the FCG experiments. Sinusoidal waveform at constant load ratio (R = 0.1)

was applied until the termination criterion of a/W = 0.7 was reached. The cyclic stress intensity factor (ΔK) was determined following the method specified in ASTM E647 [49]. The cyclic crack growth rate was evaluated by the 7-point polynomial method proposed by ASTM E647 [49].

A Zwick Roell (Ulm, Germany) 50 J miniature hammer was utilized in impact testing of 27 mm × 3 mm × 4 mm KLST specimens (60° notch angle, 1 mm depth, notch radius R = 0.1 mm, distance between anvils: 22 mm), because of limited material availability. A conversion function was established by comparison of DIN V and KLST impact energy results of solution-annealed material, following the procedure outlined by Schill et al. [50].

## 2.6. Microstructural Investigation

Specimens for the investigation of initial microstructure were electrically discharge machined (EDM) from the plate materials. For metallographic characterization the specimens were mounted in epoxy resin, ground and polished to a sub-micron finish in colloidal silica suspension and $Al_2O_3$ in dilute KOH solution for approx. 4 h and subsequently electrolytically etched at 1.5 V in 5% $H_2SO_4$ to enhance the particle/matrix contrast and to increase discriminability of the smallest (< 30 nm), very closely located particles. Characterization was accomplished by light optical (Leica MEF4, Wetzlar, Germany) and scanning electron microscopy with energy and/or wavelength dispersive X-ray spectroscopy (Zeiss Merlin, Oberkochen, Germany) / Oxford Instruments Inca/Wave (Abingdon, UK)).

## 3. Results and Discussion

### 3.1. Tensile Strength

The tensile strength of hot-rolled 17Cr1 plate material is comparatively low (cf. 17Cr1 HR in Figure 3a), when directly compared to ferritic-martensitic steel [51]. Rolling at lower temperature (i.e., below the dissolution temperature of the Laves phase) leads to a moderate rise in tensile strength in case of the cold-rolled 17Cr2 batch (cf. 17Cr2 CR in Figure 3a).

By heat treatment (cf. 17Cr2 RX + PA in Figure 3a), almost doubled ambient temperature tensile strength values are obtainable in case of the workhorse alloy. In the advanced 17Cr5 trial steel the increased W and Nb-contents cause increased tensile strength by improved solid solution hardening already in the recrystallized state (cf. Figure 3b: RX). After precipitation annealing the yield and tensile strength values, as well as rupture elongation (~15% at ambient, > 25% at high temperature), approach the range of AFM steels (cf. Figure 3b: RX + PA, exemplary microstructure in Figure 4) [52].

As a summary it can be stated, that simple precipitation annealing mainly yields strongly increased ultimate tensile and moderately improved yield strength values. If further enhanced yield strength is desired, increased dislocation density (e.g., induced by thermomechanical processing) is a measure of choice. There are obviously several effective ways to adjust the short-term mechanical properties within this alloying system. The optimum way apparently depends on the designated application, which dictates the details of material specification.

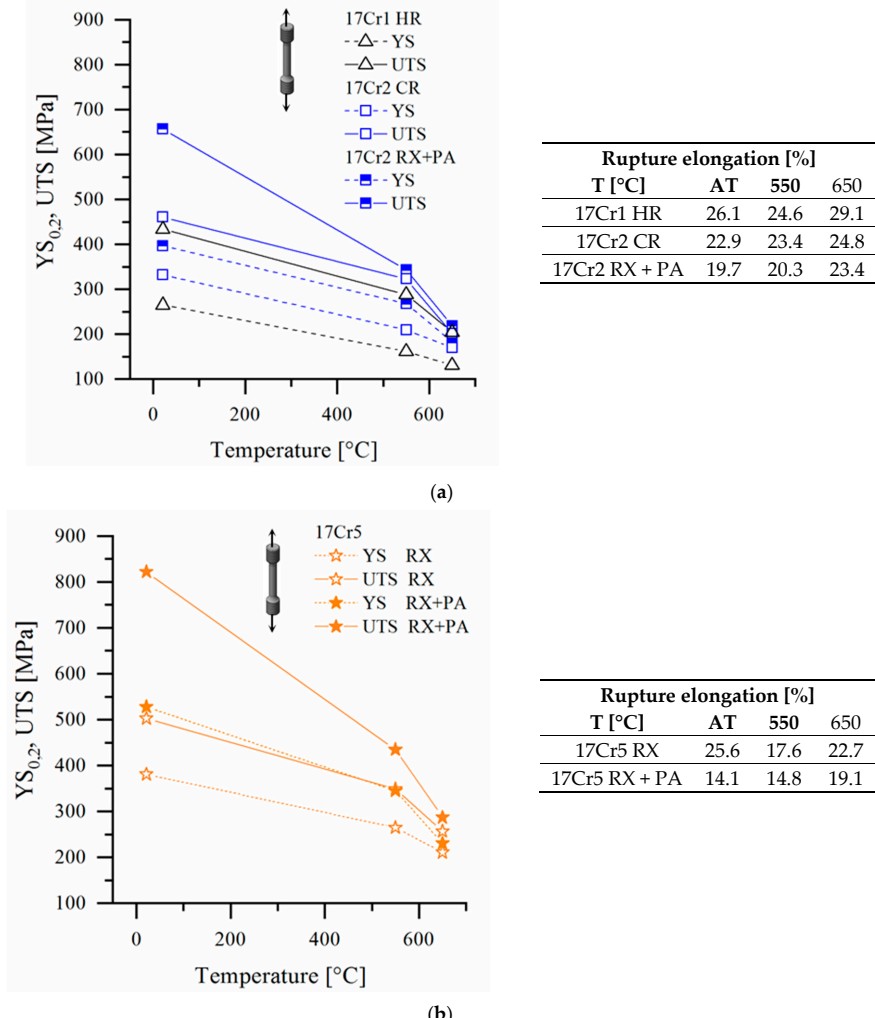

(**a**)

(**b**)

**Figure 3.** Tensile strength values of (**a**) the hot-rolled (HR) 17Cr1, the cold-rolled (CR), recrystallized and precipitation annealed (RX+PA) 17Cr2 and (**b**) the recrystallized (RX) and recrystallization and precipitation annealed (RX+PA) 17Cr5 trial steels (AT: ambient temperature).

For figure 3a table:

| Rupture elongation [%] | | | |
|---|---|---|---|
| **T [°C]** | **AT** | **550** | 650 |
| 17Cr1 HR | 26.1 | 24.6 | 29.1 |
| 17Cr2 CR | 22.9 | 23.4 | 24.8 |
| 17Cr2 RX + PA | 19.7 | 20.3 | 23.4 |

For figure 3b table:

| Rupture elongation [%] | | | |
|---|---|---|---|
| **T [°C]** | **AT** | **550** | 650 |
| 17Cr5 RX | 25.6 | 17.6 | 22.7 |
| 17Cr5 RX + PA | 14.1 | 14.8 | 19.1 |

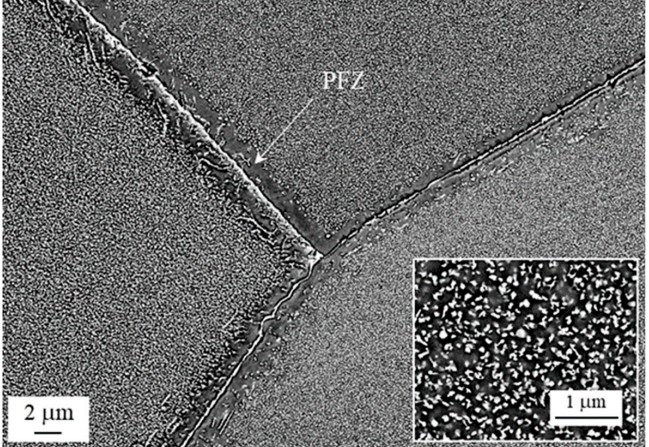

**Figure 4.** Exemplary microstructure of the recrystallized and precipitation annealed 17Cr5 (RX + PA) steel taken from a specimen head section after tensile testing (cumulated time from 600 to 650 °C: 2.75 h): Small, evenly distributed, intragranular Laves phase precipitates, almost fully covered grain boundaries with alongside formation of characteristic particle free zones (PFZ).

### 3.2. Short Crack Initiation and Growth: Technical Lifetime in Thermomechanical Fatigue Loading

The potential performance of material classes in future, flexible operation has not been considered in an adequate way by the experts yet. Thermomechanical fatigue testing can be considered as most relevant to practice, when concerning the technical lifetime of thick section components as the most critical elements in cyclic plant operation. Figure 5a displays typical thermomechanical fatigue life curves from 100% oop thermomechanical fatigue experiments. The thermal expansion of the specimens is fully obstructed in a 100 oop cycle by the testing machine, while cycling between 50 and 650 °C, causing 0.79% of mechanical strain in case of grade 92, 0.76% in case of HiperFer 17Cr2, 1.14% in case of 316L, 0.75% in case of MarBN and 0.73% in case of HiperFer 17Cr5 steel. Ferritic-martensitic steels (Figure 5a: grade 92, MarBN) exhibit high initial stress range, due to initially high intrinsic dislocation density. During an "initial phase" (approx. 300 cycles in case of grade 92, 700 cycles in case of MarBN), however, the recorded stress range drops by typically one third and the material enters a quasi-"stable" phase of slightly negative slope, which is accompanied by polygonization of the martensite lath structure [18].

Finally, the materials reach the "damage phase", which in AFM steel is characterized by a comparatively steep drop in stress range until final failure. Austenitic steel behaves the opposite way: Because of higher thermal expansion and strength at elevated temperature, it hardens in the initial phase, reaches higher stable stress range, but fails rapidly after damage initiation. Ferritic HiperFer behaves like austenitic steel in terms of cyclic hardening, but the microstructural mechanisms do differ: While in austenite initial strengthening is an effect of cyclic strain hardening, it is intensified by permanent "thermomechanically triggered precipitation" strengthening by Laves phase particles in HiperFer, which is boosted by increased dislocation density. (Cyclic) Plastic deformation not only accelerates precipitation, but furthermore refines particles (cf. Figure 6, in comparison to Figure 4).

Based on this mechanism it reaches stress ranges higher than AFM steel (17Cr2 > grade 92, 17Cr5 > MarBN), approximately on the 316L level (17Cr5) and exhibits comparatively forgiving damage and failure behavior (Figure 5a). Taking into account the outlined differences of the materials, the technical lifetime was determined from the intersection of linear approximations to the "stable" and the "damage" curve sections of the fatigue curves (cf. "$N_f$", grade 92 curve in Figure 5a). The onset of damage was assessed by deviation of the fatigue curve from a linear approximation to the stable curve section (cf. "$N_d$", grade 92 curve in Figure 5a). Relating technical lifetime $N_f$ and damage initiation $N_d$ (Table 2) to each other, demonstrates the more forgiving failure character of the HiperFer alloys, particularly of the 17Cr5 variant, concerning propagation of short cracks.

Thermo-mechanical fatigue life behavior in out-of-phase cycles of varying thermal strain obstruction (from complete, i.e., $\varepsilon_{mech.}$ = 100% to 45% of $\varepsilon_{th.}$) of the HiperFer trial alloys, ferritic-martensitic grades 92 and MarBN as well as austenitic 316L steel is depicted in Figure 5b. In comparison to grade 92, HiperFer 17Cr2 reaches almost doubled TMF lifetime over the entire span of stress (i.e., strain) ranges. With the lifetime of the recrystallized + precipitation annealed material falling close to the same main line the initial heat treatment state does not play a significant role. MarBN-type steel achieves about twice the number of loading cycles compared to the HiperFer workhorse grade (17Cr2). In the high-alloy version 17Cr5 precipitate volume fraction and thus strength and reactive hardening has been maximized. In comparison to grade 92 up to 5.5 (to MarBN 2.5 times) longer technical fatigue life is obtainable over the entire stress range, while at high stress ranges it even matches the performance of austenitic 316L (cf. Figure 5a).

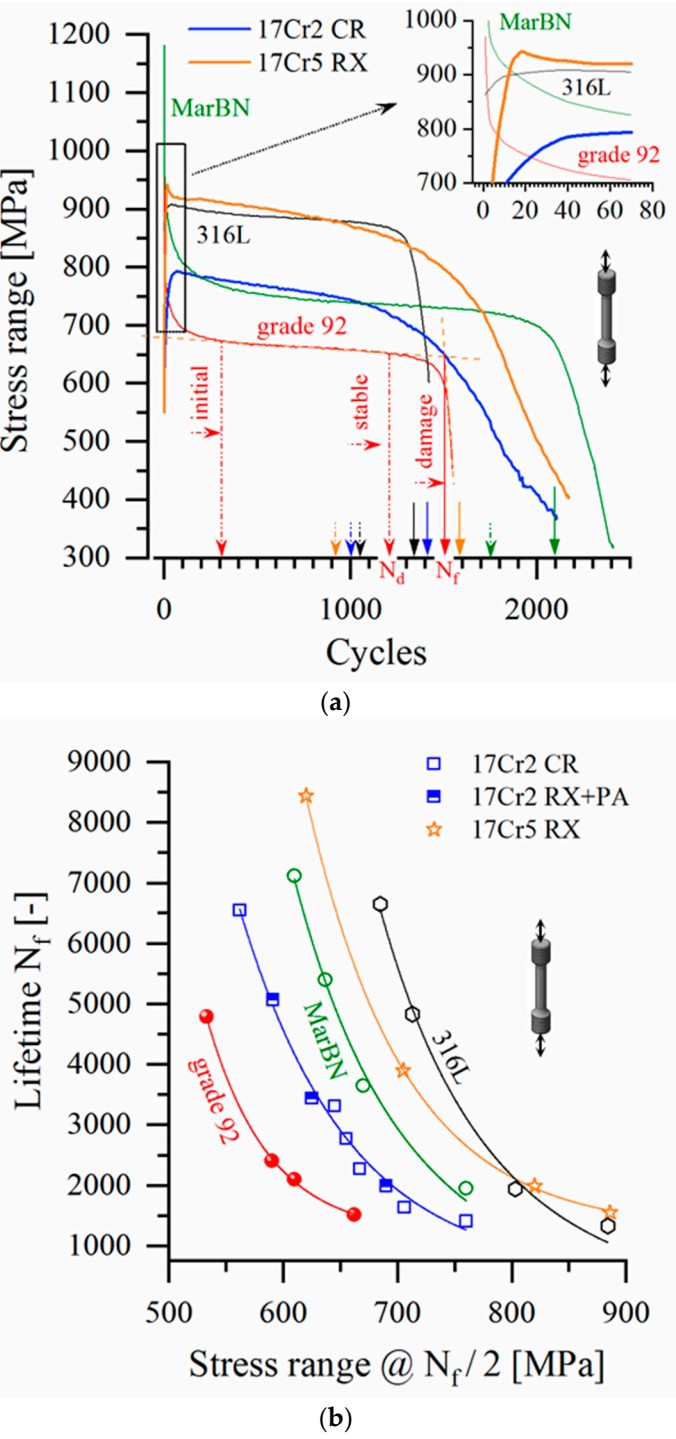

**Figure 5.** Thermo-mechanical fatigue testing results of ferritic trial, commercial ferritic-martensitic and austenitic steels (50–650 °C, $\varepsilon_{mech.}$ = −0.45 to −1$\varepsilon_{th.}$, 10 Ks$^{-1}$, no holding times at $T_{min.}$ and $T_{max.}$). (**a**) Typical fatigue life curves in 100% out-of-phase cycles ($\varepsilon_{mech.}$ = -$\varepsilon_{th.}$) and (**b**) technical lifetime over half-life stress range (i.e., as a result to differing obstruction of thermal strain, $\varepsilon_{mech.}$ = −0.45 $\varepsilon_{th.}$ to −$\varepsilon_{th.}$).

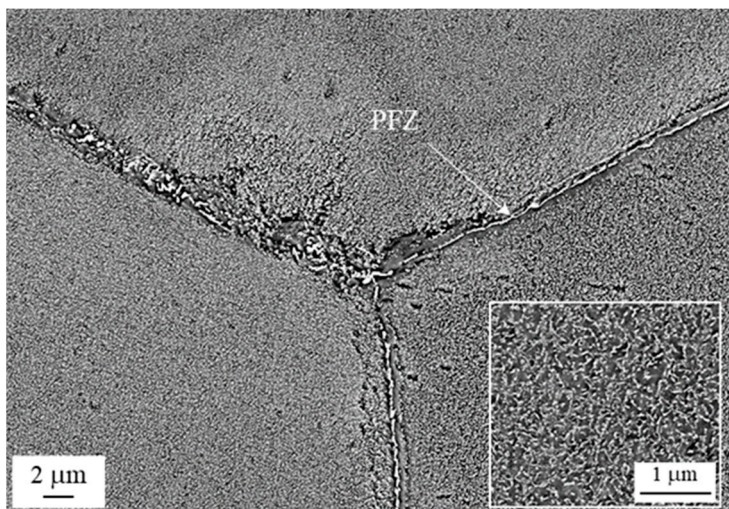

**Figure 6.** Exemplary microstructure from a 17Cr5 TMF (parameters cf. Figure 5a specimen (experiment initiated from RX state, interrupted after 1000 cycles, i.e., cumulated time from 600 to 650 °C: 2.5 h): Refined intragranular Laves phase precipitates, reduced particle free zone (PFZ) width (cf. Figure 4).

**Table 2.** Half-life stress range, technical TMF lifetime $N_f$, damage initiation $N_d$ and damage ranges (50 −650 °C, $\varepsilon_{mech.} = -\varepsilon_{th}$, 10 Ks$^{-1}$, no holding times at $T_{min.}$ and $T_{max.}$).

| Material: | $\sigma@N_{f/2}$ | $N_d$ | $N_f$ | Damage Range |
|---|---|---|---|---|
| **HiperFer 17Cr2 CR** | 760 | 974 | 1389 | 29.9% |
| **HiperFer 17Cr5 RX** | 886 | 1031 | 1569 | 34.2% |
| **Grade 92** | 662 | 1247 | 1502 | 17.0% |
| **MarBN** | 760 | 1768 | 2080 | 15.0% |
| **316L** | 884 | 1143 | 1326 | 13.8% |

*3.3. Fatigue Crack Propagation: Residual Lifetime*

In comparison to grade 92 (Figure 7) it takes 1.25 times (to 316L 1.4 times) the stress intensity to initiate propagation of pre-existing cracks in HiperFer 17Cr2 (RX + PA), while MarBN even ranges slightly higher. Nevertheless stable crack propagation is at least half an order of magnitude faster in both the AFM steels and 316L, with rising advantage of HiperFer towards increasing stress intensity. Obviously, HiperFer does not display a classical "Paris-Erdogan" [53,54] regime of positive linear proportionality of crack growth velocity (da/dN) to rising stress intensity (ΔK) in a double logarithmic representation (Figure 7). It rather reflects staircase-like curve shape with plateau regions of constant, to some extent even slightly negative proportionality (indicated by the dashed-dotted lines, inserted into the HiperFer graph in Figure 7). Crack propagation in HiperFer thus remains at a constant, sometimes even decreasing, velocity over a comparatively wide range of the cyclic crack growth curve, despite of increasing stress intensity.

The proposed reason for this behavior is a combination of "reactive precipitation strengthening", outlined in the previous section, with several other aspects: The material "dynamically" hardens, because of cyclic, plastic material distortion in front of the crack tip, what aggravates crack propagation. Another contribution stems from low-angle grain boundaries, evolving in the plastic zone in front of the crack tip (cf. sub-grain formation, caused by accumulated creep deformation, in PFZs of crept specimens, cf. Section 3.4.1, Figure 11) and rapidly becoming decorated by Laves phase particles, which frequently lead to crack branching and thus dissipation of energy away from the main crack path. Pre-existing high angle grain boundaries, covered by Laves phase particles, act in the same way.

A crack tip getting arrested at obstacles like particle clusters, (newly created) low- or (pre-existing) high-angle grain boundaries, causes increased stress intensity, in turn rising plastic distortion and consequently boosted precipitation, until a certain threshold is overcome and the crack tip breaks loose. It may further be assumed, that repetition of this mechanism causes the staircase-like curve shape in contrast to a classical Paris-Erdogan behavior. The detailed mechanism is not yet fully understood and still under investigation.

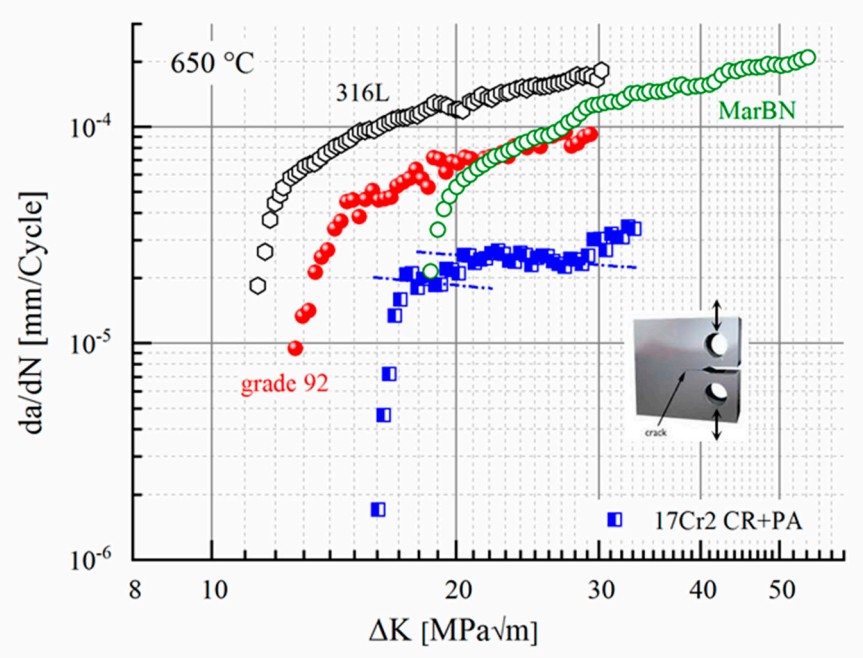

**Figure 7.** Cyclic crack growth curves (650 °C, f = 20 Hz, R = 0.1) of grade 92, MarBN, 316L and HiperFer 17Cr2 (CR + PA).

*3.4. Creep*

3.4.1. Characteristics: Low-vs. High-Alloying Variant

The hot-rolled, low-alloy variant 17Cr1 yields comparably poor creep strength at intermediate and high creep stresses (> 100 MPa; cf. 17Cr1 HR in Figures 8–10), because of low dislocation density and a lack of strengthening precipitates, when put into service in the hot-rolled state. Nevertheless, it obviously approaches the grade 92 level at practically more relevant testing stresses lower than 100 MPa (Figure 10). Cold-rolling not only yields increased tensile (cf. 17Cr2 CR in Figure 3a), but diminished minimum creep rate (Figure 9) and boosted creep rupture strength (Figures 8 and 10).

In creep, increased dislocation density has a two-fold effect in HiperFer steel: First, it accelerates precipitation and second, it sufficiently strengthens the material against creep deformation during the precipitation process. In combination, this leads to rapid establishment of a stable precipitate microstructure, reduced strain over the entire (mainly in the primary stage of) creep life (Figure 8a) and by this to a leap in creep rupture strength (Figures 8a and 10). A similar effect can be achieved by precipitation annealing of recrystallized material (cf. 100 MPa values of 17Cr2 RX + PA in Figure 8a). Compared to cold-rolling, precipitation annealing after recrystallization leads to increased primary creep strain, but in turn to a slight decrease in minimum creep rate (cf. Figure 9) and finally further improved creep rupture life (Figures 8a and 10). The higher W/Nb- contents boost solid solution and precipitation strengthening in case of the 17Cr5 composition, what in turn provides advanced creep strength. Except of increased primary stage creep strain (Figure 8b) the material performs almost comparable, no matter if put into service in the recrystallized (RX) or recrystallized + precipitation

annealed (RX + PA) state (Figures 8b, 9 and 10). Effective age-hardening characteristics thus were achieved and the design goal of 17Cr5 successfully met.

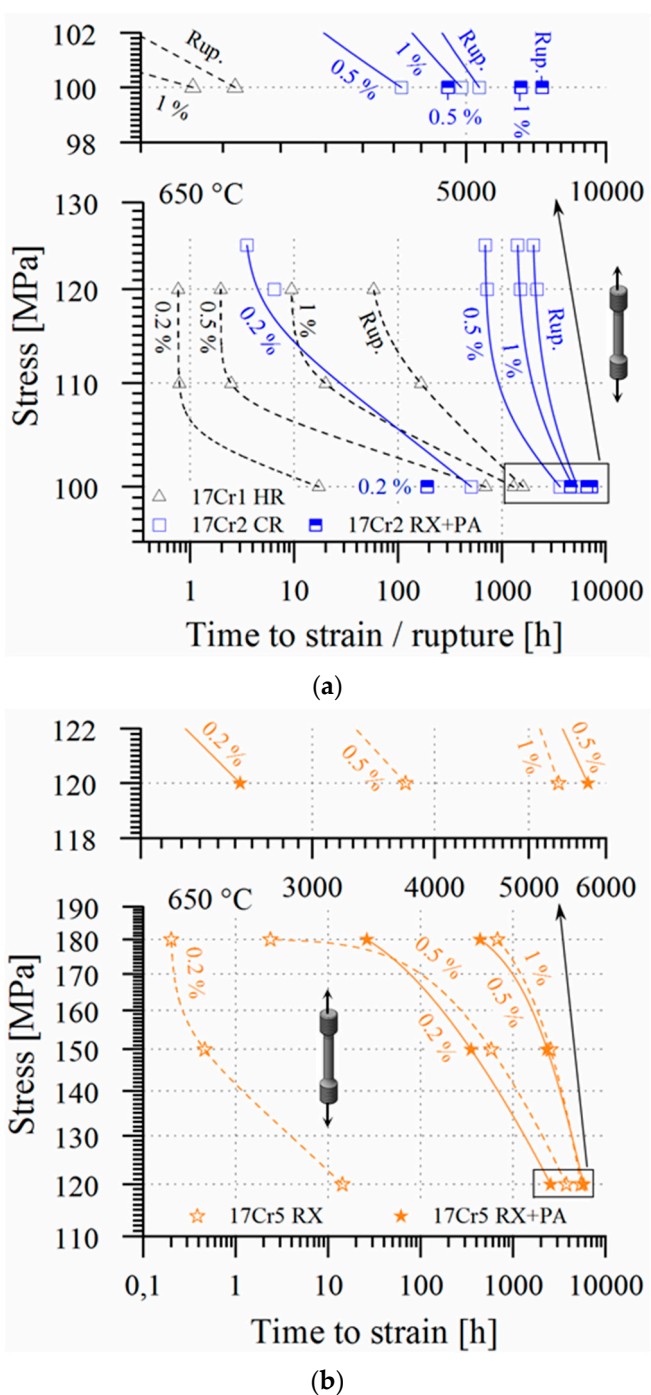

**Figure 8.** Impact of processing and heat-treatment state: Time to strain curves of hot-rolled 17Cr1, cold-rolled 17Cr2, recrystallized and precipitation annealed 17Cr2 (100 MPa data available only) (**a**), recrystallized and recrystallization and precipitation annealed 17Cr5 steel (**b**).

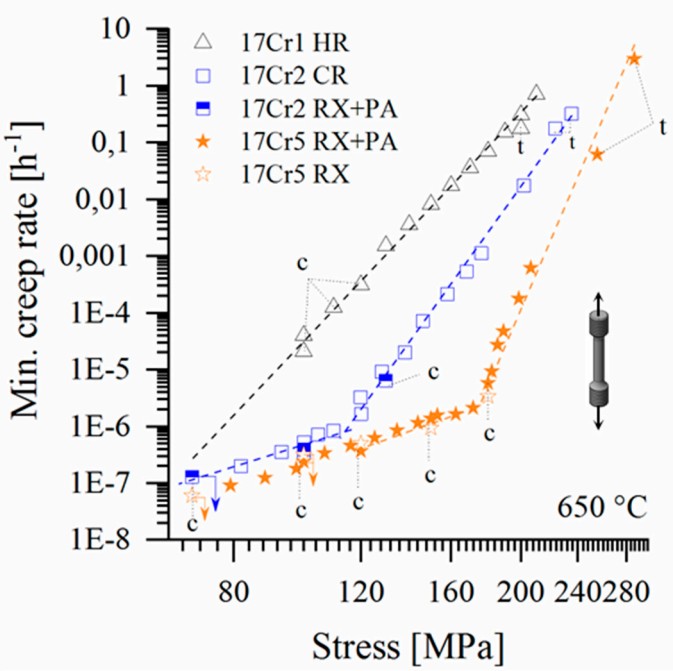

**Figure 9.** Norton-Bailey plot of the model steels in different processing and heat treatment states (650 °C; data indexed by "t"/"c" from tensile (UTS value)/uniaxial creep experiments, all others from stress relaxation testing; experiments marked with arrows are still in progress).

The change implemented in processing, from hot-rolling in case of 17Cr1 to cold-rolled 17Cr2, has a remarkable impact on the Norton-Bailey relations [55,56] of the steels (Figure 9). In comparison to the low dislocation density, hot-rolled 17Cr1 material the high dislocation density, cold-rolled 17Cr2 steel yields minimum creep rates, diminished by at least an order of magnitude over the whole stress range. The minimum creep rates of recrystallized and precipitation annealed 17Cr2 material fits well into the data of the cold-rolled steel (Figure 9). High initial dislocation density or precipitation annealing are obviously very effective in decreasing the minimum creep rates of the low alloyed model steel variant. Synopsis of the results obtained so far suggests, that the impact of thermomechanical processing (i.e., mainly initial dislocation density) may practically fade out below a stress level of approximately 70 MPa (cf. Figure 9: intersection of the extrapolated 17Cr1 HR line towards lower stress). Below this stress level (i.e., at practically relevant creep stress) / beyond this exposure time the thermomechanical treatment history is supposed to have no significant influence on creep life anymore.

### 3.4.2. Creep Rupture Strength

In Figure 10 the creep strength of the HiperFer 17Cr2 steels is ranked against various state of the art power engineering structural grades. Cold-rolled and recrystallization + precipitation annealed (RX + PA) 17Cr2 surpasses grade 92 [57] and approximately matches the performance of the novel 12 Cr steel Super VM12 [58]. While creep deformation of intermetallic particle strengthened HiperFer is controlled by growth of the strengthening Laves phase precipitates [16,47,59,60], creep damage and failure are mainly related to the accumulation of plastic deformation in PFZs (cf. Figure 11), evolving along high angle grain boundaries [36,47], in long-term application. Intragranular solid solution strengthening and particle volume fraction, as well as grain boundary particle coverage, were successfully increased and PFZ width alongside grain boundaries diminished in the 17Cr5 variant.

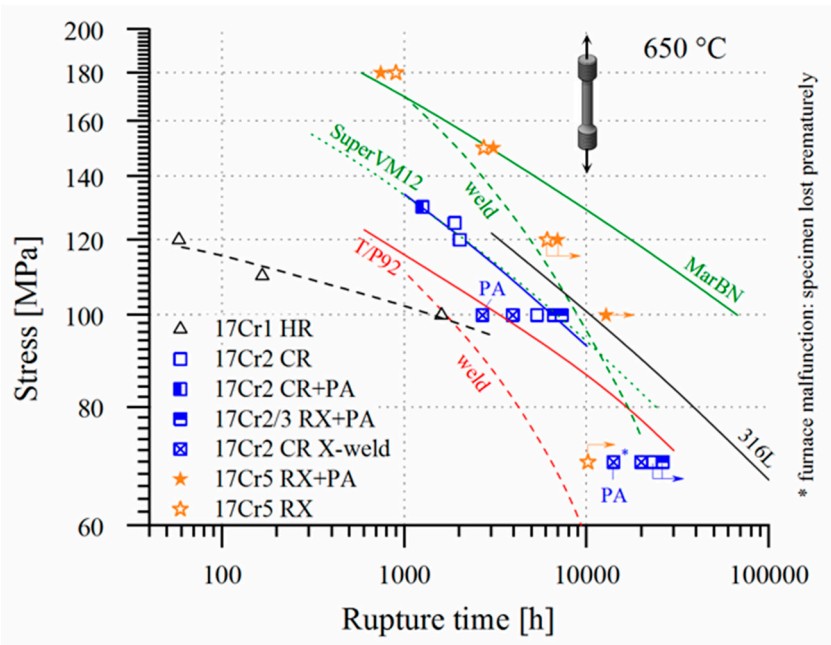

**Figure 10.** Creep rupture strength of HiperFer 17Cr1/2 and 17Cr5 in comparison to state of the art AFM and austenitic steels (100 MPa cross-weld data courtesy of Y. Yamamoto, ORNL, USA; comparative creep strength data: grade 92, 316L [57]; MarBN, mean of [61,62]; Super VM12 [58]; experiments marked with arrows are still in progress).

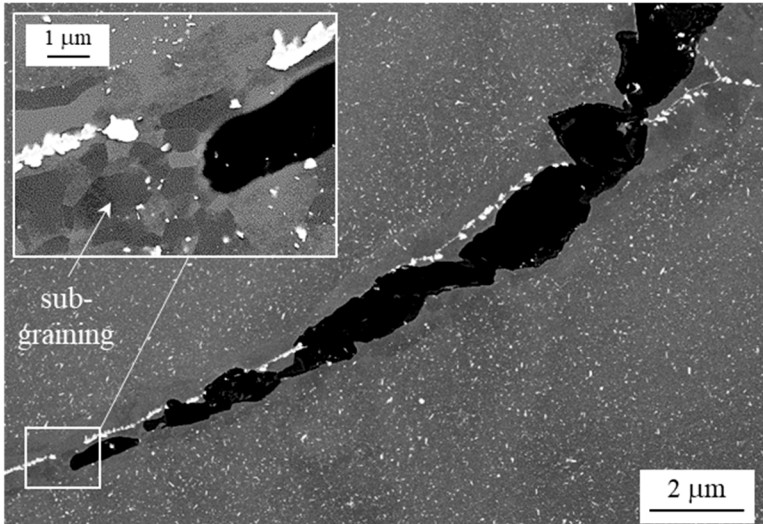

**Figure 11.** Accumulated plastic deformation, causing sub-grain formation (cf. inlay micrograph) and associated creep failure within PFZs, located at high angle grain boundaries of 17Cr5 RX material (650 °C, 150 MPa, $t_R$ = 2720 h, cf. Figure 10).

The results obtained on this improved alloy variant so far, at least at high stress (150 to 180 MPa; < 150 MPa: in progress), suggest a creep strength on the MarBN-type [61,62] level. At intermediate stresses (150 to 100 MPa, experiments still in progress) it surpasses austenitic 316L steel [57]. Nevertheless further data is necessary on lower stress levels.

First, manually produced gas tungsten arc welded (GTAW) joints of cold-rolled base material were produced, applying chemically identical filler metal strips, taken from the base metal plate. Like outlined in Section 2.3 these welds intentionally represent the worst-case of deformed (i.e., as-rolled, forged, bent,) material, being welded and put into service without conventional AFM-like post-weld

heat treatment (PWHT, i.e., annealing above application temperature). At an intermediate stress level of 100 MPa cold-rolled HiperFer 17Cr2 (CR X-weld) in the as-welded state yields rupture time slightly below, but performing simplified precipitation annealing (cf. Section 2.4) after welding pushes the weld above the grade 92 base metal level (PA in Figure 10). At low stress (70 MPa) it outplays welded grade 92 (including PWHT) by far. The precipitation annealed weld even yielded creep lifetime in the extrapolated range of MarBN-type welds. Unfortunately, the as-welded 70 MPa creep specimen was prematurely lost, because of a furnace malfunction. Creep rupture of all the 100 MPa cross-weld specimens appeared in the weld metal, while the precipitation treated 70 MPa specimen ruptured in the heat affected zone after 19879 h. The transition in fracture location is subject of further investigation.

Ease of monitoring is mandatory for high temperature structural materials. A combination of sufficient duration of tertiary creep and rupture deformation is desired. By and large the 17Cr2 (CR and RX + PA states) and 17Cr5 (RX and RX + PA states) alloys obey similar, time-modified Monkman-Grant [63] relations of time to minimum creep rate to time to rupture [64]. These indicate that more than 70% of creep life lies within the short secondary and predominantly the tertiary stage of creep. For this reason, creep life of HiperFer steel can be monitored according to the common codes of conduct.

### 3.5. Stress Relaxation

For materials utilized in bolting application resistance to stress relaxation is an important prerequisite. Figure 12 displays stress relaxation curves of the cold-rolled 17Cr2 and RX + PA 17Cr5 steels in comparison to ferritic-martensitic grade 92. A remaining stress level of 115 MPa after 600 h of relaxation at 600 °C is a benchmark value to reach for novel materials [65], when initiating stress relaxation from 250 MPa. With remaining stresses falling below 115 MPa after 7/312 h 17Cr2 CR and 17Cr5 RX + PA fail to reach this criterion in the first relaxation (Figure 12a). Nevertheless, the HiperFer grades perform remarkably better than grade 92, which drops to about 30 MPa within 87 h, while the ferritic model steels stabilize at around 95 MPa after 30 h in case of 17Cr2 CR and 468 h in case of 17Cr5 RX+PA. After reloading all the HiperFer materials perform better in the second run. 17Cr2 CR fails the criterion after another 309 h, while grade 92 after 45 h already. 17Cr5 RX + PA stabilizes well above the criterion (in a range from 150–160 MPa) for another 600 h. Thermomechanically induced precipitation of consecutive populations of Laves phase particles upon reloading is argued to be the cause for the gradual increase in performance of the HiperFer grades.

After termination of the second runs, the specimens were heated to 650 °C under the load level reached upon termination. A further relaxation run was then started after loading to 200 MPa. At 650 °C the criteria to meet are > 95 MPa after 100 h, > 55 MPa after 1000 h and > 40 MPa after 3000 h (initial stress level: 200 MPa) [65]. The grade 92 specimen fails the 100 h criterion in two consecutive runs (Figure 12b: after 4/23 h in the 3rd/4th run). In contrast to this 17Cr2 CR meets the 100 h criterion and reaches 78 MPa after another 514 h in the third run. Applying the stress relaxation rate at termination (0.0266 MPah$^{-1}$) and linearly extrapolating to 1000 h a resulting stress level of 65 MPa would have been reached and the criterion (> 55 MPa) successfully met. The third run of 17Cr5 RX + PA was terminated after 2347 h at a stress level of 70 MPa. By applying the stress relaxation rate at termination (0.0114 MPah$^{-1}$), it is apparent, that the 3000 h criterion would have been fulfilled. HiperFer can be thus considered as a potential candidate material for bolting application. Nevertheless, further investigation into heat treatment, concerning stabilization of early stage relaxation performance is needed.

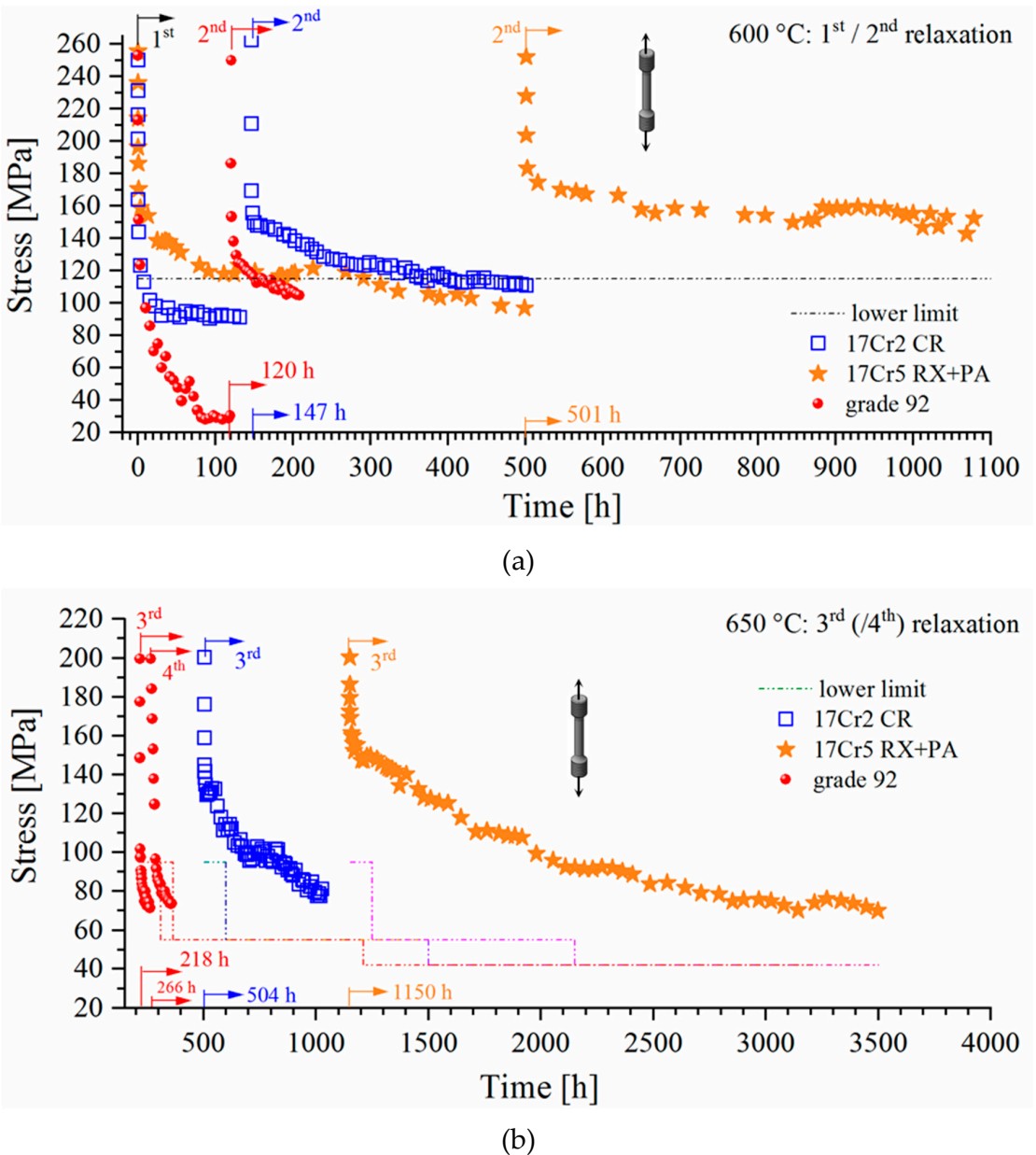

(a)

(b)

**Figure 12.** Stress relaxation of cold-rolled 17Cr2 and RX + PA 17Cr5 material in comparison to ferritic-martensitic grade 92, (**a**) 1st + 2nd run: 600 °C (initial stress level: 250 MPa) and (**b**) 3rd + (4th) run: 650 °C (initial stress level: 200 MPa).

### 3.6. Charpy Impact Strength

Relevant technical rules [66] dictate minimum impact energy values of 27/40 $Jcm^{-2}$ (longitudinal/transversal direction) at ambient temperature (indicated in Figure 13). The impact strength of ferritic, high chromium, stainless steel is generally considered problematic. If proper processing and heat treatment are applied, this can be avoided [67]. Figure 13 displays brittle to ductile transition curves of the 17Cr2 (RX + PA) and 17Cr5 (RX) variants. Recrystallized and precipitation annealed 17Cr2 (RX+PA) steel yields a steep transition curve, a DBTT of −11 °C and high upper shelf energy (all typical for quenched ferritic stainless steels [68]) and thus meets the technical specifications.

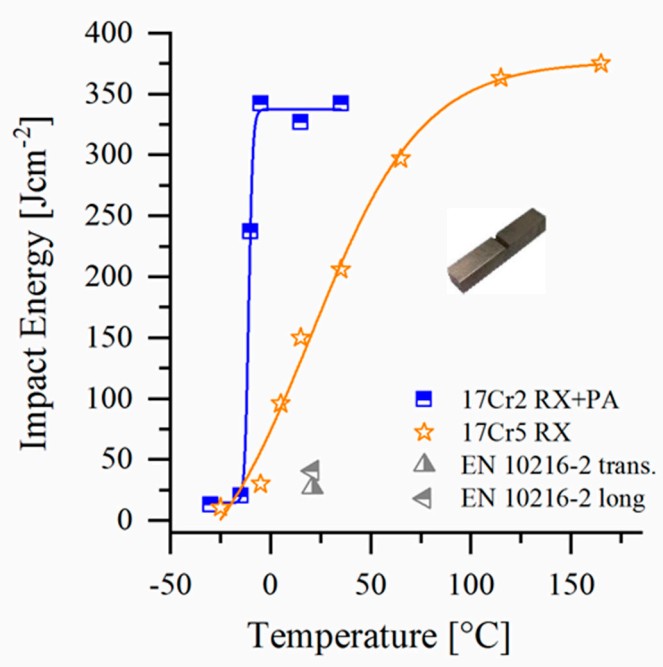

**Figure 13.** Impact strength of the recrystallized and precipitation annealed (RX + PA) 17Cr2 and recrystallized (RX) 17Cr5 steels (each data point represents the mean of three measurements).

Recrystallized 17Cr5 (RX) exhibits a less steep curve and a DBTT value of 31 °C and fulfills the requirements, too. The 17Cr5 material was recrystallized at comparatively high temperature (1100–1125 °C) and for this reason prevails in larger grain size, which typically results in a decrease in DBTT curve slope [69]. The correlations between chemical composition, processing, heat treatment, resulting microstructure and impact toughness of ferritic stainless steels are complex. For detailed information [67] may be consulted.

## 4. Conclusions

Based on its high chromium content and unique strengthening by thermally/thermomechanically induced $(Fe,Cr,Si)_2(Nb,W)$-Laves phase precipitation, HiperFer steel combines superior steam oxidation and wet corrosion resistance with improved mechanical properties. The 17Cr2 (2.5W-0.6Nb) steel combines supreme thermomechanical fatigue strength and resistance to fatigue crack propagation, promising creep / cross-weld creep and viable tensile and Charpy impact properties. The high-alloyed 17Cr5 (3.7W-1Nb) variant provides effective age-hardening capability, offers increased thermomechanical fatigue resistance and creep strength potentially combating the strongest AFM steels available. HiperFer can thus be considered as a high potential, low cost candidate material for tubing, piping, blading and bolting applications in flexibly operated, future power conversion equipment like pumped thermal electricity storage (i.e., Carnot batteries ), concentrating solar power or power-2-X conversion systems and conventional back-up power plants. Future research will concentrate on optimized processing, broadening the mechanical property database, detailed understanding and full exploitation of the "reactive hardening" mechanisms and PFZ engineering to optimize the performance of this novel steel category.

**Author Contributions:** Conceptualization, B.K., M.T. and Y.Y.; methodology, B.K., M.T. and Y.Y.; software, M.T., X.F.; validation, B.K., M.T., T.F. and J.L.B.; formal analysis, B.K., M.T., T.F., X.F., J.L.B.; investigation, B.K., M.T., Y.Y., T.F., X.F. and J.L.B.; resources, B.K. and Y.Y.; data curation, B.K., M.T., Y.Y. and T.F.; writing—original draft preparation, B.K.; writing—review and editing, Y.Y. and T.F.; visualization, B.K. and T.F.; supervision, B.K. and Y.Y.; project administration, B.K. and Y.Y.; funding acquisition, B.K. and Y.Y. All authors have read and agreed to the published version of the manuscript.

**Funding:** This research was funded by the German Ministry of Education and Research under grant number 03EK3032. Y. Yamamoto's welding research was sponsored by the U.S. Department of Energy, Office of Fossil Energy, the Crosscutting Research Program, under contract DE-AC05-00OR22725 with UT-Battele, LLC.

**Acknowledgments:** The authors would like to express their gratitude for open discussions and ideas concerning the production and processing of the HiperFer trial steels to H.-H. Dickert\*, A. Stieben\*, M. Schulte\*, W. Bleck\* and G. Hessling ((\*former affiliation) Steel Institute, RWTH Aachen University, Aachen, Germany). Supply of MarBN type steel for experimental purpose by T. U. Kern (Siemens Power and Gas) is acknowledged.

**Conflicts of Interest:** The authors declare no conflict of interest. The funders had no role in the design of the study; in the collection, analyses, or interpretation of data; in the writing of the manuscript, or in the decision to publish the results.

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
