# Peer review of "Science and Technology of High Performance Ferritic (HiperFer) Stainless Steels"

_metals, doi:10.3390/met10040463_

Round 1

Reviewer 1 Report

Firs of all: Congratulation for the Authors of the paper. The presented work describes the design and strength properties of new high performance ferritic stainless steels in a very broad way.

The work is interesting and generally very good presented, but several issues should be improved for its greater readability and usability.

The main remarks to presented work:

  1. What is on the right vertical axis on the Fig. 1?
  2. Writing using "//" is relatively unclear (first paragraph in section 2.2). Maybe it would be better to write two separate sentences?
  3. How figures 2a and 2b were obtained? There is no information about it.
  4. Generally: the use of "/" in texts is hard to read (see section 2.5, for example: “cylindrical specimens with a gauge length / diameter of 15 / 7 mm” or “the duration of the cooling / whole cycle was about 85 / 145 s”).
  5. Section 2.5 presents wide spectrum of mechanical testing. It seems advisable to present the geometry of the test samples on additional figures.
  6. Fatigue crack growth experiments: a0 is not an initial notch dept – a0 is the crack size obtained by precracking. an is the machined notch.
  7. The rupture elongation is no less important than YS and UTS. Should not be shown on the figure like these values (see Fig. 3)?
  8. There are no results of the thermomechanical fatigue testing for partly (80, 60, 45% oop) thermal expansion obstructed by the testing machine (see section 2.5 and 3.2).
  9. The data presented in the first paragraph of the section 3.2 by using “/” are unclear. Maybe they should be put in the table?
  10. Figures 5 and 8 should be larger because of their readability.
  11. How can you read the data for the εmech. = -0.45 to -1εth in Chart 5 (see description of this figure)?
  12. There are no curves for 17Cr2/3 RX-PA data on Fig. 8a – only points. Why?

Conclusion:

In my opinion, the paper is very interesting for journal readers and suitable for publication in it after minor corrections.

Author Response

Thank you very much and kind regards

Bernd Kuhn

Reviewer 2 Report

The authors describe the development and evaluation of high-Cr fully ferritic steel alloys. The use of English language and grammar is generally good, but it should be review for readability.

The description of the “working horse” alloy and then subsequent description of the heats produced and evaluated here is a bit hard to follow. Suggested to replace “working horse” with “workhorse” or “standard”. Were 17Cr2 and 17Cr3 actually the same heat, if so why the double notation?

Line 128: any reason why the higher temperature is first? (1180 – 1140)

Line 128-130 – following the processing in this sentence is difficult with the “//” notation

Line 131 “_HR” and Line 134 “_CR” notation is defined and then never used, why bother

Lines 148-151: Add more detail to inform the readers of this processing was performed by the authors or if the material was supplied in this condition

Line 286-289: parenthetical i.e. is quite long and should be constructed as a full sentence

Section 4 “Discussion” is more of a summary than discussion and should be renamed. In light of this, the authors may want to consider expanding some of the results subsections to include more of the discussion of the results. The work done and reported in this manuscript was well done, but the readability of the paper makes it difficult to parse it out.

The proposed superior steam oxidation and wet corrosion resistance is not back-up in this manuscript, only mentioned to be assumed based on the equilibrium predicted phases. If these studies have not been conducted on this alloy series (not referenced) then they should be conducted in future work to make sure there are no surprises.

Much of the results focus on fatigue and creep related properties, with CVN and tensile thrown in as a reference point. As such, the authors may want to consider also revising their title and/or abstract to reflect the body of work that the manuscript details. The discussion of these implication of the measured properties (fatigue/creep) in these alloys could also be further expanded.

Author Response

(The authors gave the same response as above.)

Reviewer 3 Report

I think the text specially in introduction and experimental part is written more like a report than a scientific paper. 
on the other hand, since the application is mainly for high temperature it is interesting to compare the behavior of the new designed materials at different temperatures.

Author Response

(The authors gave the same response as above.)

Round 2

Reviewer 3 Report

Congratulations!